# Pituitary-Gland-Based Genes Participates in Intrauterine Growth Restriction in Piglets

**DOI:** 10.3390/genes13112141

**Published:** 2022-11-17

**Authors:** Xiang Ji, Qi Shen, Pingxian Wu, Hongyue Chen, Shujie Wang, Dong Chen, Yang Yu, Zongyi Guo, Jinyong Wang, Guoqing Tang

**Affiliations:** 1College of Animal Science and Technology, Sichuan Agricultural University, Chengdu 611130, China; 2Chongqing Academy of Animal Sciences, Chongqing 402460, China; 3National Center of Technology Innovation for Pigs, Chongqing 402460, China; 4Chongqing Animal Husbandry and Technology Diffusion Centre, Chongqing 401120, China

**Keywords:** IUGR, Rongchang pig, pituitary, transcriptomics

## Abstract

Intrauterine growth restriction (IUGR) is a major problem associated with piglet growth performance. The incidence of IUGR is widespread in Rongchang pigs. The pituitary gland is important for regulating growth and metabolism, and research has identified genes associated with growth and development. The pituitary gland of newborn piglets with normal birth weight (NBW group, *n* = 3) and (IUGR group, *n* = 3) was collected for transcriptome analysis. A total of 323 differentially expression genes (DEGs) were identified (|log2(fold-change)| > 1 and *q* value < 0.05), of which 223 were upregulated and 100 were downregulated. Gene Ontology (GO) functional and Kyoto Encyclopedia of Genes and Genomes (KEGG) pathway enrichment analyses showed that the DEGs were mainly related to the extracellular matrix, regulation of the multicellular organismal process, tissue development and angiogenesis, which participate in the growth and immune response in IUGR piglets. Moreover, 7 DEGs including *IGF2*, *THBS1*, *ITGA1*, *ITGA8*, *EPSTI1*, *FOSB*, and *UCP2* were associated with growth and immune response. Furthermore, based on the interaction network analysis of the DEGs, two genes, *IGF2* and *THBS1*, participated in cell proliferation, embryonic development and angiogenesis. *IGF2* and *THBS1* were also the main genes participating in the IUGR. This study identified the core genes involved in IUGR in piglets and provided a reference for exploring the effect of the pituitary gland on piglet growth.

## 1. Introduction

Intrauterine growth restriction (IUGR) is the poor growth of the embryo development of mammals during pregnancy, resulting in a neonatal birth weight 10% below the average weight for the same gestational age. IUGR is a serious disorder associated with compromised fetal development during pregnancy [1]. Crowded conditions in the womb and placental insufficiency have been implicated in impaired fetal or postnatal growth and development [2]. As multiparous mammals, pigs are often affected by IUGR [3]. In addition, IUGR piglets are associated with high morbidity and mortality, stunted growth, and poor quality carcasses during growth and development [2], which may emerge in an asymmetrical (70–80% of cases) or a symmetrical (20–30% of cases) manner [4]. Several useful measures have been proposed to reduce the negative effects of IUGR piglets, in which the impact is minimized by feed nutrition [5,6,7] and environmental management. However, the genetic mechanisms of IUGR piglets are not well defined, which hinders the growth of IUGR piglets.

Growth traits constitute a significant indicator for the pig farming industry and are of great concern for the development of the livestock industry. IUGR is a widespread disease related with high mortality and morbidity in both human beings and animals, especially in piglets, which represents great financial losses in the pig farming industry. In recent decades, with the continuous development of genetic improvement and reproductive technology, the number of litters farrowed by a single sow has significantly increased, which is an important factor contributing to IUGR. Fetal growth (increase in cell number or individual mass) and development (cell differentiation or functional changes in tissue structure) are complex physiological processes influenced by genetics, epigenetics, maternal maturity, and other factors [8]. Compared to normal birth weight (NBW), IUGR can negatively affect nutrient absorption [9], muscle development [10], intestinal health [11] and immune resistance [12] of piglets. Thus, rearing IUGR piglets is likely to impact more negatively on the pig farming industry.

The pituitary gland is located at the base of the brain, regulates the growth and reproduction of animals, and affects the endocrine function by mediating the transmission of hypothalamic signals to target organs. The mature pituitary gland consists of adenohypophysis and neurohypophysis. The hormones secreted by adenohypophysis mainly regulate the growth and development of the body, maintain metabolic balance, and regulate the body’s ability to withstand stress [13]. The pituitary gland plays a significant role in the endocrine system by secreting hormones that regulate normal physiological functions of the body. Transcriptome profiling studies have implicated intrauterine adverse environments and the disruption of the hypothalamus–pituitary–adrenal (HPA) axis in the development of IUGR [14]. However, the underlying mechanisms associated with pituitary disfunction in IUGR piglet remain poorly understood.

Comparative analysis of the pituitary transcriptome of IUGR and NBW piglets has demonstrated the role of the pituitary in the occurrence of IUGR in human beings. High-throughput methods have been widely used to elucidate both physiological and pathological states caused by IUGR in various species [15,16,17]. Transcriptome, also known as RNA sequencing (RNA-seq), is a genome-wide technique that enables the exploration of differences in specific gene types and expression levels [18]. With the continuous advancements in this technology, significant improvements have been made in pig genetic breeding. Studying the genetic mechanism affecting IUGR traits at the molecular level and screening candidate genes could deepen our understanding of the genetic mechanisms underlying the occurrence and effects of IUGR. Reports on the genes that participate in the occurrence of IUGR piglets remain scanty. Thus, considering that the pituitary gland is required for the regulation the growth of the organism, the aim of the present study was to compare the pituitary transcriptome of IUGR and NBW piglets to identify genes related to growth and development. Moreover, Protein–Protein Interaction (PPI) Networks regulated by the DEGs were identified based on data in the STRING database. From the findings described herein, the hub genes affecting the emergence of IUGR in piglets were identified.

## 2. Materials and Methods

### 2.1. Animals and Sampling

All experiments involving animal handing and sample collection described herein were previously approved by and in compliance with the requirements of the Institutional Animal Care and Use Committee, College of Animal Science and Technology, Sichuan Agricultural University, Sichuan Province, China (No. DKY20170913). All piglets were obtained from Shuanghe Farm, located in the Rongchang district, Chongqing China. Among piglets, three male piglets with the highest body weight and three male piglets with the lowest body weight constituted, respectively, the NBW group and the IUGR group. All piglets were sacrificed immediately after birth, and fractions of the pituitary gland of each animal were collected, immediately frozen in liquid nitrogen, and then stored at −80 °C until RNA isolation.

### 2.2. RNA Extraction and Purification

Total RNA from pituitary tissue was isolated using the TRIzol Plus RNA Purification Kit (Invitrogen, Waltham, MA, USA) according to the manufacturer’s instructions. RNA purity was determined by Nano Photometer spectrophotometer (IMPLEN, Westlake Village, CA, USA) using OD260/280 and OD260/230 ratios. RNA was quantified using Nano Drop (Thermo Fisher Scientific, Waltham, MA, USA). Moreover, Agilent RNA Nano 6000 Kit for the Agilent Bioanalyzer 2100 system (Agilent Technologies, Santa Clara, CA, USA) was used to assess the integrity of RNA samples.

### 2.3. Library Construction and Sequencing

All RNA samples used to construct the pituitary transcriptome cDNA sequencing libraries had an RNA integrity number (RIN) of 7.5 or higher, in line with the sequencing quality requirements. The results of RNA integrity are shown in Appendix A. RNA libraries were constructed using NEB Next Ultra^TM^ RNA Library Prep Kit for Illumina (NEB, Ipswich, MA, USA) following the manufacturer’s instructions. After quality control, the RNA library was pooled according to its effective concentration and the desired data volume, and the libraries were sequenced in an Illumina second-generation high-throughput sequencing platform to generate 150 bp paired-end reads.

### 2.4. Quality Control of Sequencing Data and Comparison with Reference Genomes

Raw reads were first filtered using in-house Perl scripts. Reads with more than N bases and more than 50% low-quality bases (Q < 20) with sequencing junctions were removed using Trimmomatic software. Finally, after the clean reads were obtained, sequencing data statistics revealed that Q20 > 95%, Q30 > 93% indicated good sequencing quality, thus confirming that the data could be used for subsequent comparison and data analysis. Clean reads were aligned to the pig genome available in public databases (Sscrofa 11.1).

### 2.5. Differential Gene Expression Analysis

Differential gene expression analysis was performed based on read count data obtained from gene expression analysis. For samples with biological duplicates, differential gene expression analysis of the NBW and IUGR groups was performed using the DESeq R package (www.huber.embl.de/users/anders/DESeq/ accessed on 4 July 2022). DESeq enables determining differential expression for gene expression data using a model based on negative binomial distribution [19]. *p*-values were adjusted by Benjamini and Hochberg’s method for controlling the false discovery rate (FDR). Significant differential gene expression was set at *p* < 0.05.

### 2.6. Enrichment Analysis of DEGs

Gene Ontology enrichment analysis (http://www.geneontology.org/ accessed on 5 July 2022) was performed using the GO seq software for analysis [20], whose algorithm is based on the Wallenius non-central hyper-geometric distribution. By estimating the preferred gene length, it is possible to calculate the probability that a differentially expressed gene is enriched by the probability of the GO term more accurately.

Kyoto Encyclopedia of Genes and Genomes (KEGG) (http://www.kegg.jp accessed on 5 July 2022), which is a systematic analysis of gene function, genomic information database, helps researchers to study gene and expression information as an overall network.

### 2.7. Detection of Gene Expression

The expression of the DEGs was analyzed by RT-qPCR using the CFX384 multiplex real-time fluorescence quantitative PCR instrument (Bio-Rad, Berkeley, CA, USA) platform. First-strand cDNA was synthesized from 300 ng of total RNA using Script cDNA synthesis kit (Bio-Rad, Hercules, CA, USA) in a 4 μL total volume. The resulting cDNA was ten-fold diluted and used in 20μL RT-qPCR mixture as follows: Power SYBP Green Master Mix, 10 μL; gene-specific upstream and downstream primers (10 μmol/L), 0.5 μL; sterile water, 8 μL; and cDNA template, 1 μL. The total reaction volume was 20 μL. The RT-qPCR reaction conditions were as follows: initial denaturation at 95 °C for 1 min, subsequent denaturation of 40 cycles of 95 °C for 15 s and annealing at 63 °C for 25 s. Each sample was evaluated independently three times. Relative gene expression analysis for each gene was determined using the 2^−△△Ct^ method. ACTB was the internal reference gene. All primers were designed using Primer-BLAST in the NCBI database (https://www.ncbi.nim.nih.gov/tools/primer-blast/ accessed on 1 October 2022). Data on the expression of genes of interest are shown in Appendix A.

### 2.8. Statistical Analysis

The results were presented as the mean ± standard error of the mean (SEM). All statistical analyses were performed using SPSS software, version 23 (IBM, Armonk, NYC, USA). Differences between the IUGR and NBW groups were analyzed by unpaired two-tailed Student’s *t*-test, and statistical significance was set at *p* > 0.05.

## 3. Results

### 3.1. Statistical Analysis

In the present study, body weight measurement of piglets is summarized in Table 1. All piglets included in the study were male and originated from different sows. Piglets were divided into NBW and IUGR groups according to their body weight. Piglets were sacrificed at birth to minimize interference by external factors. The average body weight of the NBW group was 0.77 ± 0.02 Kg, and of the IUGR group it was 0.46 ± 0.02 Kg. The body weight of piglets in the NBW group was approximately 1.7 times higher than that of piglets in the IUGR group, which was significantly different (Figure 1). The weight of all three IUGR piglets was within two standard deviations in relation to the average body weight of normal-sized piglets, in line with the established estimates.

### 3.2. Pituitary Transcriptome Sequencing and Reads Mapping

A total of 49.41, 55.88, 52.30, 54.20, 54.16, and 56.28 million raw reads were obtained from pituitary tissue samples of IUGR1, IUGR2, IUGR3, NBW1, NBW2, and NBW3, respectively. To determine differences in mRNA expression levels between the pituitary gland of piglets of different body weights, RNA-seq data were obtained from the pituitary gland of piglets in the NBW and IUGR sample groups. The average number of original reads for the six sample groups was 53,704,980. After discarding low-quality reads, 84.75–89.00% of raw reads were considered as high-quality reads, of which 83.52–87.79% was mapped uniquely to the pig reference genome, whereas 1.16–1.35% showed multiple matches (hereafter ‘good matches’ referred to the unique and multiple matches). The Q30 value is the percentage of bases for which the recognition accuracy exceeds 99.9%. The average Q30 value in this study was 94.02% (Table 2). To determine intergroup difference and intragroup sample duplication, principal component analysis (PCA) was performed on all samples. Collectively, the PCA results showed intergroup isolation and intragroup aggregation (Appendix A).Therefore, RNA-seq data were consistent with subsequent data analysis.

### 3.3. Expression Characterization in Pituitary Transcriptome

Using *p* < 0.05 and |log-2fold change| > 1 as threshold criteria, a total of 323 DEGs were identified, of which 223 were upregulated and 100 were downregulated in the pituitary transcriptome of piglets in the NBW and IUGR sample groups. All identified DEGs are shown in Appendix A. Each gene annotated as a DEG is shown as a dot in Figure 2a. The top ten upregulated genes and the top five downregulated genes are shown in Table 3. The number of upregulated DEGs was higher than the number of downregulated in IUGR compared with NBW, and significantly altered DEGs are indicated as red and green dots. Moreover, the identification of DEGs in the pituitary gland of piglets in the NBW and IUGR sample groups was performed by clustering analysis of DEGs. As shown in Figure 2b, both high and low weight piglets clustered together, which demonstrated that the obtained data were reliably accurate. Among DEGs analyzed by RNA-seq, extracellular matrix (ECM), tissue development and angiogenesis were closely related to IUGR occurrence and were associated with development stages.

### 3.4. Functional Annotation and Pathway Enrichment Analysis of DEGs

To further explain the function of the 323 identified DEGs, GO and KEGG pathway enrichment analyses were performed. The GO terms are grouped into three categories: biological processes, cellular components, and molecular functions; 172 terms (*p* < 0.05) were significantly enriched in these three categories. The top 30 terms (including 20 terms for biological processes, 9 terms for cellular components and 1 term for molecular functions) were further analyzed to determine the associated regulatory network as shown in Figure 3a. Within the GO term biological processes, most enriched genes were found within regulation of multicellular organismal process, regulation of developmental process, and tissue development, which are genes known to be related to the growth and development of the organism, implying that they have an impact on IUGR piglets. Within the GO term cellular component, most enriched genes were found in extracellular regions, with more than 22% of candidate genes being annotated within this term. In addition, the 323 identified DEGs were evaluated based on the KEGG pathway database, and 12 pathways (*p* < 0.05) were found enriched in pituitary transcriptomes of piglets in the NBW and IUGR groups (Figure 3b). Among the enriched pathways, the PI3K−Akt signaling pathway, folate biosynthesis, protein digestion, and absorption and glycerolipid metabolism are knowingly related to growth and development in pigs.

### 3.5. PPI Network of the DEGs

The STRING database (https://strig-db.org accessed on 9 July 2022) was used to construct the PPI network diagram of DEGs, and the Cytoscape software (version 3.9.1 https://cytoscape.org/ accessed on 10 July 2022) was used to identify the most important nodes, whose results are shown in Figure 4. The darker colors in the picture indicate hub genes *IGF2* and *THBS1*, which may participate on the occurrence of IUGR in piglets.

### 3.6. Validation of RNA-seq Results Using RT–PCR

To verify the RNA-seq results, eight genes among DEGs were randomly selected to validate their expression in the NBW and IUGR experimental groups (*n* = 3), namely *FOSB*, *IGF2*, *EPSTI1*, *SLC7A1*, *UCP2*, *THBS1*, *ITGA1*, and *ITGA8* using RT-qPCR analysis. The results of the validation of RNA-seq results are shown in Figure 5. Among these, the *SLC7A1* and *THBS1* were not detected due to the fact that their expression was too low (Ct > 35). The expression trends of the other selected genes were consistent with the transcriptome sequencing results. In addition, the expression of *FOSB*, *ITGA1*, and *ITGA8* showed significant (*p* < 0.05) differences between the NBW and IUGR groups. Collectively, these results validate the reliability of the RNA-seq data.

## 4. Discussion

The pituitary gland is a neuroendocrine organ that regulates several important physiological functions, including growth, sexual development and reproduction, metabolism, lactation, and immune response [21]. The neuroendocrine centers of the organism mainly comprise the hypothalamus, pituitary gland, and pineal gland [22]. The pituitary gland mainly secretes hormones for growth and development of piglets. At present, certain genes and active substances have been suggested to be involved in growth performance in pigs and are closely related to the hypothalamic-pituitary function. Therefore, the stability of the pituitary function has an important role in maintaining the body’s metabolism [23]. It has been shown that IUGR individuals have a potential compensatory mechanism for rapid weight gain in the late growth phase [6]. Although compensatory growth is considered a positive aspect, there is growing evidence of severe metabolic disorders in newborns with IUGR. Moreover, the compensatory growth of newborns with IUGR is mainly reflected in the increase in fat rather than lean muscle [24], which may contribute to insulin resistance and obesity, and combined may lead to an increased risk of diseases in the late growth phase of the organism. However, the specific role of the pituitary gland in these physiological activities is not well understood. The present study is the first to investigate the pituitary gland as the main organ affecting the emergence of IUGR piglets, thus providing a reference for the early prevention of IUGR in piglets.

Integrins are integral cell surface proteins involved in the adhesion, migration, and proliferation of cells. Mchugh et al. described that alterations in integrin gene expression were involved in multiple disease processes, including inflammation, cardiovascular disease, and cancer [25]. In vertebrates, 24 integrin-encoding genes have been described, of which 16 of them encode integrin α subunits, while the remaining 8 encode integrin β subunits [26]. It is known that the interaction of integrins with the extracellular matrix is important for cell function and survival [27]. In addition, integrins also act on osteoblasts to promote osteoclast differentiation [28]. In the present study, two members of the integrin family were identified, *ITGA1*, *ITGA8*, among significantly DEGs in the NBW and IUGR groups. Considering the GO enrichment analysis, these genes were significantly enriched in the cell surface receptor signaling pathways and extracellular regions, suggesting that integrins may affect IUGR development via the cell surface receptor signaling pathways.

Immune function is an important indicator of body health. IUGR causes atrophy and alters the function of the thymus and other organs in newborn animals [29]. Moreover, lymphocyte migration, leukocyte-cell adhesion, and chemotaxis regulation have been described during the immune response of IUGR piglets and are associated with immune dysregulation [30]. *EPSTI1* is an IL-28A-mediated interferon-inducible gene that mediates antiviral activity via the RNA-dependent protein kinase (PKR) genes [31]. Moreover, the *EPSTI1* gene may play a potential regulatory role in the body’s natural immunity, being shown to promote tumor invasion and metastasis in various cancer types [32]. In the present study, *EPSTI1* was upregulated in IUGR piglets, suggesting its possible involvement in immune function. However, only a few studies have attempted to elucidate the role of *EPSTI1* on the growth and development of piglets. Plasma glucose concentration in IUGR piglets was lower than that of normal piglets, which also implies that energy utilization and glucose absorption in IUGR piglets might not be as efficient as that in normal piglets [33]. As part of the mitochondrial uncoupling protein family, *UCP2* plays an important role in energy regulation and is a negative regulator of insulin secretion [34]. Herein, *UCP2* was found to be upregulated in IUGR piglets, suggesting that compared to NBW, IUGR animals are at a higher risk of developing diabetes [35].

The insulin-like growth factor 2 (IGF2), encoded by the *IGF2* gene, regulates fetal growth, development, and metabolism, playing an important role in embryonic development [36]. Liu et al. found that upregulated *IGF2* expression induces phosphorylation and subsequent activation of the PI3K/Akt signaling pathway, promoting protein translation, embryonic development, and cell proliferation [37]. As a peptide hormone, *IGF2* is highly homologous to insulinogen. Lien et al. have confirmed that during the newborn period, IUGR pups exhibited impaired insulin secretion and reduced vascularity [38]. Thus, these results imply that *IGF2* might be an important gene affecting IUGR piglets, consistent with our findings. Thrombospondin1 (THBS1) was the first identified endogenous inhibitor of angiogenesis [39]. *THBS1* encodes a type of matricellular protein that is both an extracellular matrix and a cell surface receptor [40]. A previous study showed that *THBS1* negatively affects angiogenesis and cell growth [41]. It has been shown that *THBS1* could inhibit skeletal muscle capillary angiogenesis since the knockdown of *THBS1* in mice led to increased muscle capillaries and enhanced motility [42]. In addition, tissue development in patients with liver cancer has been attributed to the high expression of *THBS1*, which promotes angiogenesis [43]. Taken together, these findings reveal the importance of the *THBS1* gene in organismal angiogenesis. In the present study, *THBS1* was upregulated in IUGR piglets, which was in line with the findings of previous studies. *FOSB* belongs to the *FOS* gene family, which has a significant function in bone formation [44]. In the present study, the pituitary gland was found to have a possible effect on the production of skeletal tissue in the body, and skeletal muscle is one of the major insulin-targeted tissues for whole body maintenance. Moreover, in skeletal physiology, there is growing evidence that natural truncated variants formed by *FOSB* selective shearing might have an important function in bone formation [45]. Eagle et al. found that *FOSB* overexpression decreases the excitability of pyramidal neurons, whereas inhibition of *FOSB* increases neuronal excitability [46].

## 5. Conclusions

Herein, transcriptomic analysis of the pituitary gland of NBW and IUGR piglets was conducted, and DEGs affecting growth and development in IUGR in piglets have been identified. Moreover, PPI network analysis of DEGs was performed using the STRING database, which identified *THBS1* and *IGF2* as the main genes affecting the occurrence of IUGR in piglets. In addition, cell surface receptor signaling, tube morphogenesis, and angiogenesis development and regulation were the pathways found to play a significant role in the occurrence of IUGR piglets. Collectively, this study addresses the knowledge gap in the role of pituitary tissue in regulating IUGR emergence in piglets, laying the foundation for further research on genes regulating IUGR occurrence.

## Figures and Tables

**Figure 1 genes-13-02141-f001:**
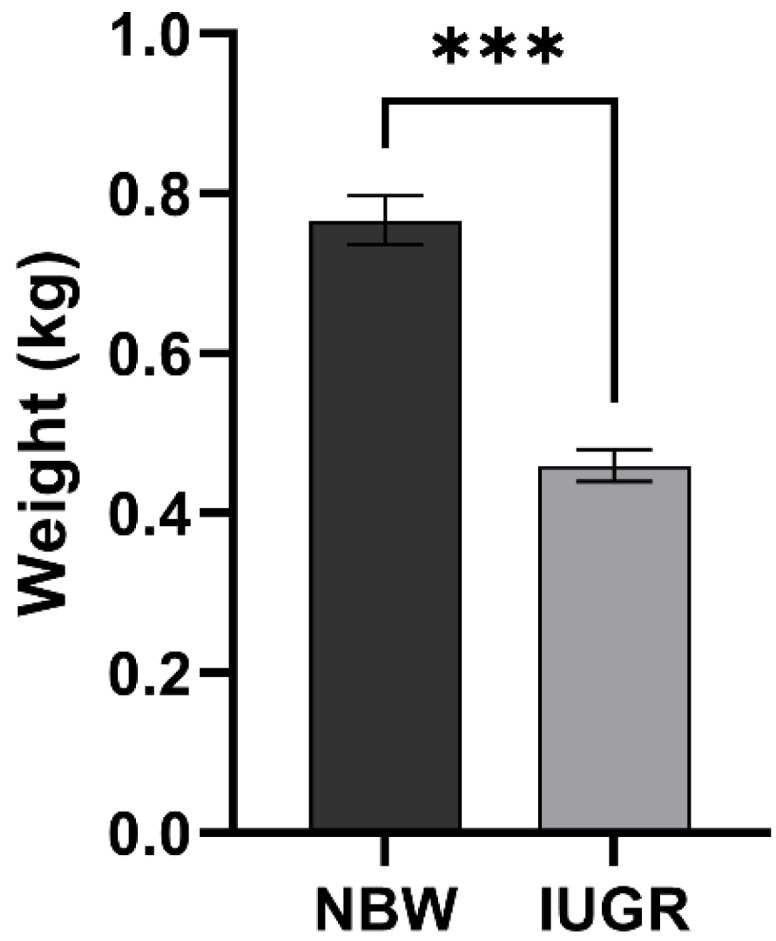
The mean body weight of piglets in the NBW and IUGR groups (*n* = 3). Data are shown as mean ± standard deviation (SD), “***” indicates significant differences (*p* < 0.001).

**Figure 2 genes-13-02141-f002:**
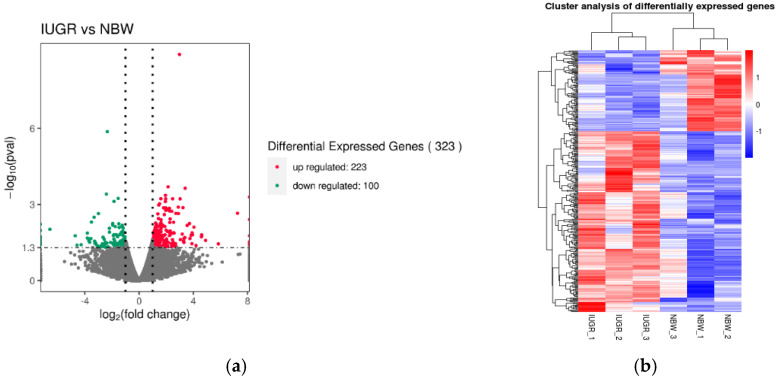
DEGs Analysis. (**a**) Volcano plot of total expression of genes in both the IUGR and NBW groups. The *x*-axis represents the log2 (fold-change) values for gene expression, and the *y*-axis repression the −log10 (*p*-value) (padj in the figure represents the *p*-value; *p* < 0.05 indicates −log10(padj) > 1.3). (**b**) The color scale represents Fragments Per Kilobase of exon model per Million mapped fragment- (FPKM-) normalized log10 (transformed counts). The horizontal bars represent genes. Each vertical column represents a sample, and sample names are as indicated. Red indicates upregulated genes, while blue indicates downregulated genes.

**Figure 3 genes-13-02141-f003:**
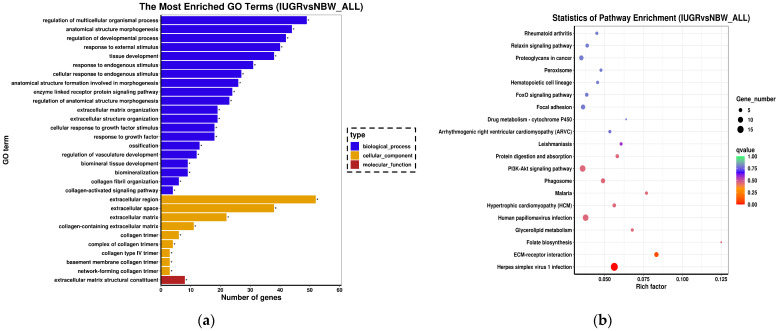
Enrichment Analysis. (**a**) GO term enrichment analysis of DEGs, with the 30 top terms obtained by GO enrichment. The *x*-axis represents the number of DEGs in the term, and the *y*-axis represents the enriched GO terms. The top 30 GO terms included 20 terms for biological process, 9 terms for cellular component and 1 term for molecular function. “*” indicates *p* < 0.05. (**b**) Enriched KEGG pathways (top 20) for the DEGs that were obtained by KEGG enrichment. The size of the dots indicates the number of expressed genes in the pathway, and the color of the dots represents the different Q value ranges of the pathway.

**Figure 4 genes-13-02141-f004:**
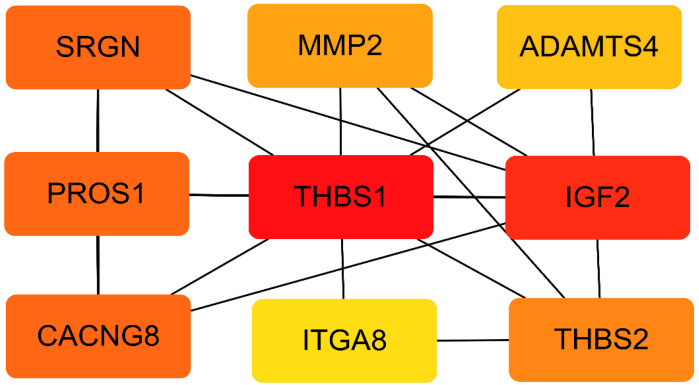
Identification of hub genes in the protein–protein interaction (PPI) network. Cytoscape’s plug-in CytoHubba uses the MCC algorithm to select the most important module from the PPI network. The darker the color, the higher the gene relatedness.

**Figure 5 genes-13-02141-f005:**
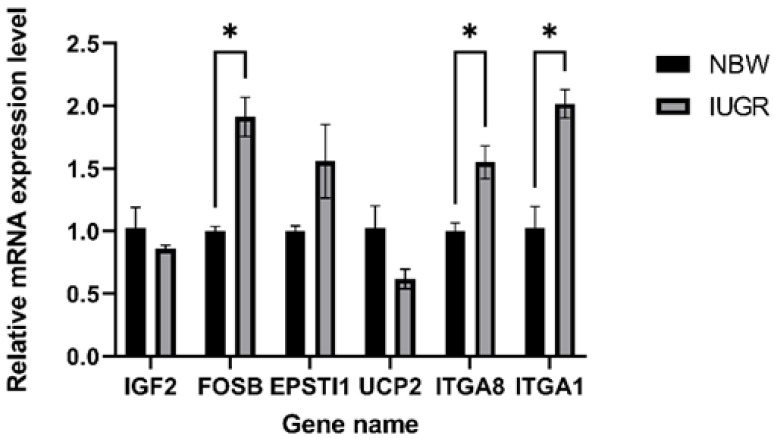
Validation of 6 DEGs between the NBW and IUGR piglet pituitary by RT-qPCR. The black columns indicate the NBW results; the gray columns indicate the IUGR results. The *x*-axis represents the names of the DEGs, and the *y*-axis represents the log2 fold-change from RT-qPCR. “*” indicates *p* < 0.05.

**Table 1 genes-13-02141-t001:** Statistical Analysis of piglet weight in two groups.

Group	Sample ID	Body Weight (kg)	Mean of Litter Weight (kg)	SD	Number of Litter	Number of Alive Litter
NBW	NBW1	0.76	0.68	0.09	12	10
	NBW2	0.80	0.69	0.10	14	11
	NBW3	0.74	0.67	0.09	11	9
IUGR	IUGR1	0.44	0.69	0.09	12	11
	IUGR2	0.48	0.70	0.10	10	9
	IUGR3	0.46	0.68	0.08	13	10

Each piglet in both groups was from a different sow. The average weight of the NBW group was 0.77 kg, while that of the IUGR group was 0.46 kg. All IUGR piglets were 1.5 standard deviations (SD) below the mean litter weight and met the definition of the IUGR piglets.

**Table 2 genes-13-02141-t002:** Summary of reads and matches.

Sample Name	Raw Reads	Clean Reads	Total Mapped	Multiple Mapped	Uniquely Mapped	Q30 (%)
IUGR1	49,413,660	49,049,218	42,890,986	567,530	42,323,456	93.77
IUGR2	55,880,062	55,364,908	46,921,638	679,036	46,242,602	93.67
IUGR3	52,301,590	51,877,426	45,486,562	644,576	44,841,986	93.75
NBW1	54,197,428	53,609,282	47,021,570	700,368	46,321,202	94.50
NBW2	54,158,458	53,521,394	47,632,404	643,780	46,988,624	93.97
NBW3	56,278,682	55,660,888	47,666,648	751,188	46,915,460	94.46

**Table 3 genes-13-02141-t003:** Top ten upregulated and top five downregulated in the IUGR group relative to the NBW group.

Gene ID	Gene Name	Log2 Fold-Change	*p*-Value
ENSSSCG00000007805	ATP2A1	−3.32	0.00319
ENSSSCG00000025535	TFAP2B	−2.19	0.00869
ENSSSCG00000001036	TFAP2A	−1.95	0.00654
ENSSSCG00000014833	UCP2	−1.54	0.00058
ENSSSCG00000036814	CLEC11A	−1.50	0.00576
ENSSSCG00000010727	GPR26	1.51	0.00094
ENSSSCG00000004464	ITGA1	1.61	0.00060
ENSSSCG00000011973	COL8A1	1.94	0.00043
ENSSSCG00000005498	PAPPA	1.94	0.00076
ENSSSCG00000029311	MYPN	2.23	0.00141
ENSSSCG00000015268	FMO1	2.25	0.00181
ENSSSCG00000024514	SHISAL2B	2.89	0.00127
ENSSSCG00000031616	FOSB	2.98	1.22 × 10^−9^
ENSSSCG00000003259	OSCAR	3.24	0.00134
ENSSSCG00000037572	EPSTI1	3.40	0.00023

## Data Availability

All the raw sequence data of RNA-seq have been deposited in the National Genomics Data Center (NGDC) Genome Sequence Archive (GSA) under BioProject accession numbers: CRA007844.

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
