# Peer review of "Pituitary-Gland-Based Genes Participates in Intrauterine Growth Restriction in Piglets"

_genes, 2022, doi:10.3390/genes13112141_

Round 1
Reviewer 1 Report
The study demonstrates the “Pituitary Transcriptome Profiling Identified Genes Involved in 2 Intrauterine Growth Restricted Piglets in Rongchang Pigs”. The current study employed current omics technology in Pigs. Although the main subject of the study is potentially interesting and worth publishing, authors should clearly explain the contribution of their work when compared to other previous relevant studies on the subject.
Grammar and writing style needs improving throughout, sentences are often difficult to follow due to poor structure, incorrect words used (out of context) or missing words. I have commented specifically on this in a few places throughout the manuscript.
Line 87-97: please clearly explain design of animal study. How did you group animals into NBW and IUGR? Did the author only use males? If so, why did you obtain female?
Line 102-103: Please provide the results for RNA integrity from the bioanalyzer as a supplemental file
Line 126-127: Please clearly explain the FDR value which was set to identify differentially expressed genes
Reviewer 2 Report
The manuscript describes the analysis of the pituitary transcriptome from Rongchang piglets in order to detect genes involved in intrauterine growth restriction. The study is very innovative and complete, including gene ontology functional and KEGG pathway enrichment analyses. Figures, tables and supplemental material are useful for better comprehension of results. About reference list, 14/46 (30.4%) references are from the last four years; therefore, references are updated.
Suggestions:
Line 17: it is said KEGG. This abbreviation should be explained.
Lines 130-131: It is said: Gene Ontology enrichment analysis (http://www.geneontology.org/) was conducted useing the software GO seq for analysis[20]. Whoes algorithm is based on the Wallenius non-central hyper-geometric distribution. “Using” should be changed by using. The period after the first sentence should be replaced by a comma.
Lines 135-137: it is said: KEGG (Kyoto Encyclopedia of Genes and Genomes, http://www.kegg.jp), which is a systematic analysis of gene function, genomic information database, which helps researchers to study gene and expression information as an overall network. Maybe the second “which” would be deleted.
Lines 139-140: it is said : RT-qPCR analysis of DEGs in a CFX384 multiplex real-time fluorescence quantitative PCR instrument (OLABO, USA). There is no verb in his sentence.
Lines 260-262: it is said: To verify RNA-seq results, eight genes among DEGs were randomly selectrd to validate …. “Selectrd” should be written selected.
Lines 334-335: It is said; In the present study, the pituitary gland was found to have a possible effect on the production of skeleta tissue in the body.. “Skeleta” should be written skeletal.
In Table 1 (Statistical Analysis of piglet weight in two groups). The column SD should follow the column that indicates the average weight of the litter.
